# Prognostic Impact of Histologic Subtype and Divergent Differentiation in Patients with Metastatic Urothelial Carcinoma Treated with Enfortumab Vedotin: A Multicenter Retrospective Study

Akinori Minato [1,*], Nobuki Furubayashi [2], Yujiro Nagata [1], Toshihisa Tomoda [3], Hiroyuki Masaoka [4], Yoohyun Song [4], Yoshifumi Hori [5], Keijiro Kiyoshima [6], Takahito Negishi [2], Kentaro Kuroiwa [5], Narihito Seki [4], Ikko Tomisaki [1], Kenichi Harada [1], Motonobu Nakamura [2] and Naohiro Fujimoto [1]

[1] Department of Urology, School of Medicine, University of Occupational and Environmental Health, Kitakyushu 807-8555, Japan; harada311@med.uoeh-u.ac.jp (K.H.); n-fuji@med.uoeh-u.ac.jp (N.F.)
[2] Department of Urology, National Hospital Organization Kyushu Cancer Center, Fukuoka 811-1395, Japan; furubayashi.nobuki.yc@mail.hosp.go.jp (N.F.)
[3] Department of Urology, Oita Prefectural Hospital, Oita 870-8511, Japan
[4] Department of Urology, Kyushu Central Hospital of the Mutual Aid Association of Public School Teachers, Fukuoka 815-8588, Japan
[5] Department of Urology, Miyazaki Prefectural Miyazaki Hospital, Miyazaki 880-8510, Japan
[6] Department of Urology, Japanese Red Cross Fukuoka Hospital, Fukuoka 815-8555, Japan
* Correspondence: a-minato@med.uoeh-u.ac.jp

**Abstract:** Subtype of urothelial carcinoma (SUC), defined here as urothelial carcinoma with any histologic subtype or divergent differentiation, is a clinically aggressive disease. However, the efficacy of enfortumab vedotin (EV) against SUC remains unclear. Hence, this study aimed to assess the oncological outcomes of patients with SUC treated with EV for metastatic disease. We retrospectively evaluated consecutive patients with advanced lower and upper urinary tract cancer who received EV after platinum-based chemotherapy and immune checkpoint blockade therapy at six institutions. The objective response rate (ORR), progression-free survival (PFS), and overall survival (OS) were compared between patients with pure urothelial carcinoma (PUC) and those with SUC. We identified 44 and 18 patients with PUC and SUC, respectively. Squamous differentiation was the most common subtype element, followed by glandular differentiation and sarcomatoid subtype. Although patients with SUC had a comparable ORR to those with PUC, the duration of response for SUC was short. Patients with SUC had poorer PFS than those with PUC; however, no significant difference was observed in OS. Multivariate analysis revealed that SUC was significantly associated with shorter PFS. Although the response of metastatic SUC to EV was similar to that of PUC, SUC showed faster progression than PUC.

**Keywords:** enfortumab vedotin; antibody–drug conjugate; histologic subtype; metastatic urothelial carcinoma; prognosis; survival; divergent differentiation; variant histology

## 1. Introduction

Urothelial carcinoma (UC) is a common malignancy of the genitourinary system, typically affecting the bladder, renal pelvis, and the ureter. However, bladder cancer accounts for the majority of the UC cases. In 2020, 573,200 new cases of bladder cancer and 212,500 related deaths were recorded worldwide according to the GLOBOCAN estimates [1]. In recent years, immune checkpoint inhibitors (ICIs) have become the standard treatment for advanced UC as an adjuvant, maintenance, or second-line therapy after platinum-based chemotherapy [2–4]. However, most cases of advanced UC tend to progress. In the EV-301 phase 3 clinical trial, enfortumab vedotin (EV), which is an antibody–drug

conjugate (ADC) directed against nectin-4, improved the survival of patients after platinum-based chemotherapy and ICI therapy failure [5]. Thus, in November 2021 in Japan, EV monotherapy was approved as a third-line regimen for patients with advanced UC. In 2023, we reported our early experience with the high effectiveness and tolerability of EV monotherapy for metastatic UC in a single-center cohort [6].

Pure conventional UC (PUC) is a predominant histology of upper or lower urinary tract malignancies, and it also has different morphologic categories within a tumor type. Owing to improved pathologic recognition [7], a subtype of UC (SUC), defined here as UC with any histologic subtype or divergent differentiation, has been frequently observed [8]. Thus, we previously reported that SUC accounted for approximately 31% of muscle-invasive bladder cancers, 12% of upper tract urinary cancers, and 34% of metastatic diseases [9]. SUC generally presents with an already advanced stage at diagnosis, indicating an aggressive biological behavior [10]. Recent studies have focused on the morphologic category in UC as a prognosticator in patients with locally advanced bladder cancer [11] and upper urinary tract cancer [12]. Ultimately, the results regarding prognosis in the surgical setting for patients with SUC are conflicting. Additionally, evidence of survival from systemic therapy for patients with metastatic SUC is insufficient [13,14] as the efficacy of EV on SUC has rarely been reported, thereby remaining unclear. Therefore, the therapeutic effect of EV in patients with SUC must be investigated further.

This study aimed to assess the oncological outcomes of patients with metastatic SUC who received EV monotherapy in real-world clinical practice from the Uro-Oncology Group in Kyushu (UROKYU) study population.

## 2. Materials and Methods

### 2.1. Patient Population

We retrospectively reviewed 63 consecutive patients with histologically confirmed metastatic PUC or SUC in the upper or lower urinary tract who had received EV after chemotherapy and immunotherapy at six institutions between December 2021 and August 2023, using the UROKYU study population. In this study, SUC was defined as the mixed presence of UC and a histologic subtype or divergent differentiation based on the World Health Organization Classification of Tumors [8]. The histomorphological subgroup was determined based on reports provided by dedicated pathologists at each institution without a central review. Prior radical surgeries included cystectomy and nephroureterectomy. In our cohort, no patient underwent metastasectomy or cytoreductive surgery. Some patients were diagnosed with PUC or SUC solely based on a small biopsy specimen for primary or metastatic tumor sites, including transurethral resection of bladder tumors and uretero-scopic biopsy of the renal pelvis or the ureter. All patients showed radiologically confirmed disease progression after undergoing platinum-based chemotherapy and subsequent ICI therapy for metastatic disease. After excluding one patient for whom the therapeutic effect of EV could not be evaluated, we included 62 patients for the analysis. Our study protocol was approved by the University of Occupational and Environmental Health Institutional Review Board (approval no. CRG23-017) and the ethics committee of each institution.

### 2.2. Patient Management

EV was administered intravenously at a dose of 1.25 mg/kg on days 1, 8, and 15, and the cycle was repeated every 28 days until disease progression, unacceptable adverse events, or consent withdrawal occurred. No upper limit was put on the number of cycles in the case of no progression. Moreover, EV treatment was stopped immediately in the case of unacceptable adverse events or consent withdrawal. Follow-up evaluation included physical examination, laboratory tests, and chest–abdominal–pelvic computed tomography, which was performed at baseline and after every two to three cycles of EV. If symptoms appeared, appropriate additional examinations were conducted.

### 2.3. Evaluation

Tumor response was assessed as the best response according to the Response Evaluation Criteria in Solid Tumors, version 1.1. [15]. We defined objective response rate (ORR) as the proportion of patients with a complete response (CR) or a partial response (PR), and the disease control rate as the proportion of patients with CR, PR, or stable disease (SD) without progressive disease (PD). The duration of response was the time from PR or CR onset until progression or death from any cause, whichever was earlier.

Moreover, progression-free survival (PFS) was calculated from the date of EV introduction to the date of disease progression or death, whichever occurred earlier, or to the last follow-up in patients without disease progression. Overall survival (OS) was calculated from the date of administration to the date of death from any cause or to the last follow-up in patients who survived.

### 2.4. Statistical Analysis

All statistical data were analyzed using EZR ver. 1.40 (Easy R; Saitama Medical Center, Jichi Medical University, Saitama, Japan), which is a graphical user interface for R (The R Foundation for Statistical Computing, Vienna, Austria) [16]. Between-group differences were assessed using Fisher's exact test for categorical variables and Mann–Whitney $U$ test for continuous variables. Response duration, PFS, and OS were estimated using the Kaplan–Meier method and compared using the log-rank test. For the univariate and multivariate analyses of clinicopathological factors, we used the Cox proportional hazard models. The significant factors influencing the cause of progression or death as the dependent outcome were recruited in multivariate analyses. A $p$ value less than 0.05 was considered statistically significant.

## 3. Results

### 3.1. Patient Characteristics

Of the 62 patients enrolled, 44 (71%) and 18 (29%) had PUC and SUC, respectively. The most common subtype element was squamous differentiation (16.1%), followed by glandular differentiation (6.5%) and sarcomatoid subtype (3.2%) (Table 1).

**Table 1.** Histologic type of patients treated with enfortumab vedotin.

| Histologic Type | Number of Patients (%) | Primary Tumor Site Lower/Upper Urinary Tract |
|---|---|---|
| PUC | 44 (71) | 20/24 |
| SUC | 18 (29) | 12/6 |
| SUC subgroup | | |
|   Squamous differentiation | 10 (16.1) | 6/4 |
|   Glandular differentiation | 4 (6.5) | 4/0 |
|   Sarcomatoid subtype | 2 (3.2) | 0/2 |
|   Plasmacytoid subtype | 1 (1.6) | 1/0 |
|   Neuroendocrine differentiation | 1 (1.6) | 1/0 |

Abbreviations: PUC, pure urothelial carcinoma; SUC, subtype of urothelial carcinoma.

Table 2 compares the baseline characteristics of patients with PUC and SUC. Age, sex, Eastern Cooperative Oncology Group performance status (ECOG-PS), primary tumor site, anemia occurrence, proportion of liver metastasis, and prior treatment pattern did not significantly differ between the two patient groups. Furthermore, the dosing cycles of EV administered were comparable between the groups. Withdrawal of EV resulting in adverse events and consent withdrawal occurred in five patients and one patient, respectively. Regardless of their severity, we did not note any deaths caused by adverse events during the EV treatment.

**Table 2.** Patient characteristics.

| | PUC (*n* = 44) | SUC (*n* = 18) | *p* Value |
|---|---|---|---|
| Age, median (IQR) | 73 (68–76) | 73 (71–79) | 0.415 |
| Sex, *n* (%) | | | 0.355 |
| Male | 31 (70.5) | 15 (83.3) | |
| Female | 13 (29.5) | 3 (16.7) | |
| ECOG-PS score, *n* (%) | | | 0.667 |
| 0 | 19 (43.2) | 7 (38.9) | |
| 1 | 18 (40.9) | 8 (44.4) | |
| 2 | 4 (9.1) | 3 (16.7) | |
| 3 | 3 (6.8) | 0 (0) | |
| Primary tumor site, *n* (%) | | | 0.166 |
| Lower urinary tract | 20 (45.5) | 12 (66.7) | |
| Upper urinary tract | 24 (54.5) | 6 (33.3) | |
| Histologic diagnosis, *n* (%) | | | 0.412 |
| Prior radical surgery | 24 (54.5) | 12 (66.7) | |
| Biopsy | 20 (45.5) | 6 (33.3) | |
| Anemia (Hb < 10 g/dL), *n* (%) | 14 (31.8) | 7 (38.9) | 0.768 |
| Liver metastasis, n (%) | 9 (20.5) | 4 (22.2) | 1.00 |
| Number of prior lines of therapy, *n* (%) | | | 0.842 |
| 2 | 32 (72.7) | 13 (72.2) | |
| ≥3 | 12 (27.3) | 5 (27.8) | |
| Prior immune checkpoint blockade, *n* (%) | | | 1.00 |
| Anti-PD-1 | 28 (63.6) | 11 (61.1) | |
| Anti-PD-L1 | 16 (36.4) | 7 (38.9) | |
| EV cycles, median (IQR) | 4 (3–7) | 4 (2–7) | 0.365 |

Abbreviations: IQR, interquartile range; ECOG-PS, Eastern Cooperative Oncology Group performance status; Hb, hemoglobin; PD-1, programmed cell death protein 1; PD-L1, programmed death-ligand 1; EV, enfortumab vedotin; PUC, pure urothelial carcinoma; SUC, subtype of urothelial carcinoma.

### 3.2. Response

Table 3 shows the best overall response in the PUC and SUC groups. The ORR and disease control rate were similar between the two groups. The detailed response in patients with SUC was as follows: squamous differentiation (PR in eight and PD in two patients); glandular differentiation (SD in two and PD in two patients); sarcomatoid subtype (PD in two patients); plasmacytoid subtype (PD in one patient); and neuroendocrine differentiation (SD in one patient). Among the 33 patients evaluated as PR or CR, the response duration for patients with SUC was significantly shorter than for patients with PUC (median, 3.7 months vs. 7.3 months, *p* = 0.003) (Figure 1).

**Table 3.** Observed efficacy of enfortumab vedotin stratified by histologic type.

| | PUC (*n* = 44) | SUC (*n* = 18) | *p* Value |
|---|---|---|---|
| Response, *n* (%) | | | 0.475 |
| CR | 3 (6.8) | 0 (0) | |
| PR | 22 (50.0) | 8 (44.4) | |
| SD | 10 (22.7) | 3 (16.7) | |
| PD | 9 (20.5) | 7 (38.9) | |
| Objective response rate (CR + PR), *n* (%) | 25 (56.8) | 8 (44.4) | 0.413 |
| Disease control rate (CR + PR + SD), *n* (%) | 35 (79.5) | 11 (61.1) | 0.20 |

Abbreviations: CR, complete response; PR, partial response; SD, stable disease; PD, progressive disease; PUC, pure urothelial carcinoma; SUC, subtype of urothelial carcinoma.

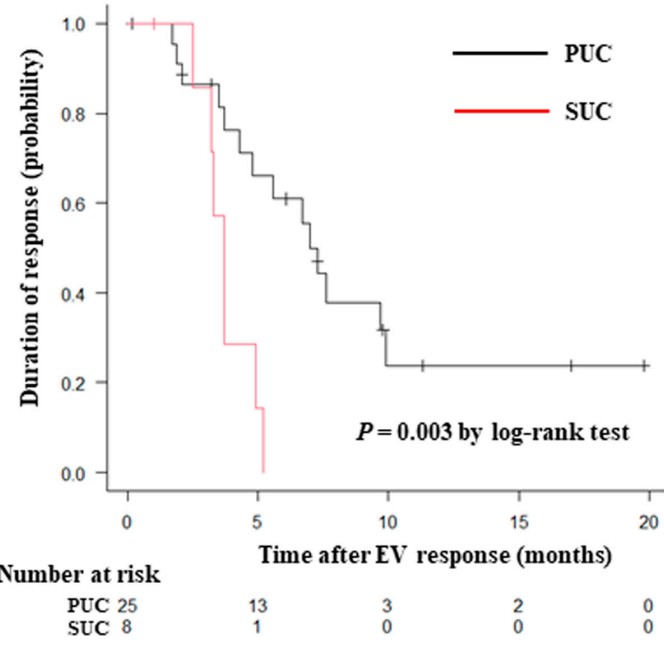

**Figure 1.** Kaplan–Meier curves for the duration of response (complete or partial responders) to enfortumab vedotin in patients with PUC and SUC. Abbreviations: PUC, pure urothelial carcinoma; SUC, subtype of urothelial carcinoma.

*3.3. Survival*

The median follow-up time was 7.1 months (interquartile range, 4.0–11.8), during which 47 (75.8%) patients experienced progression and 30 (48.4%) patients died. PFS was poorer in patients with SUC than in those with PUC (median, 4.2 months vs. 5.9 months; $p = 0.045$) (Figure 2a), whereas OS showed no significant difference between such groups (median, 7.3 months vs. 16.1 months; $p = 0.065$) (Figure 2b).

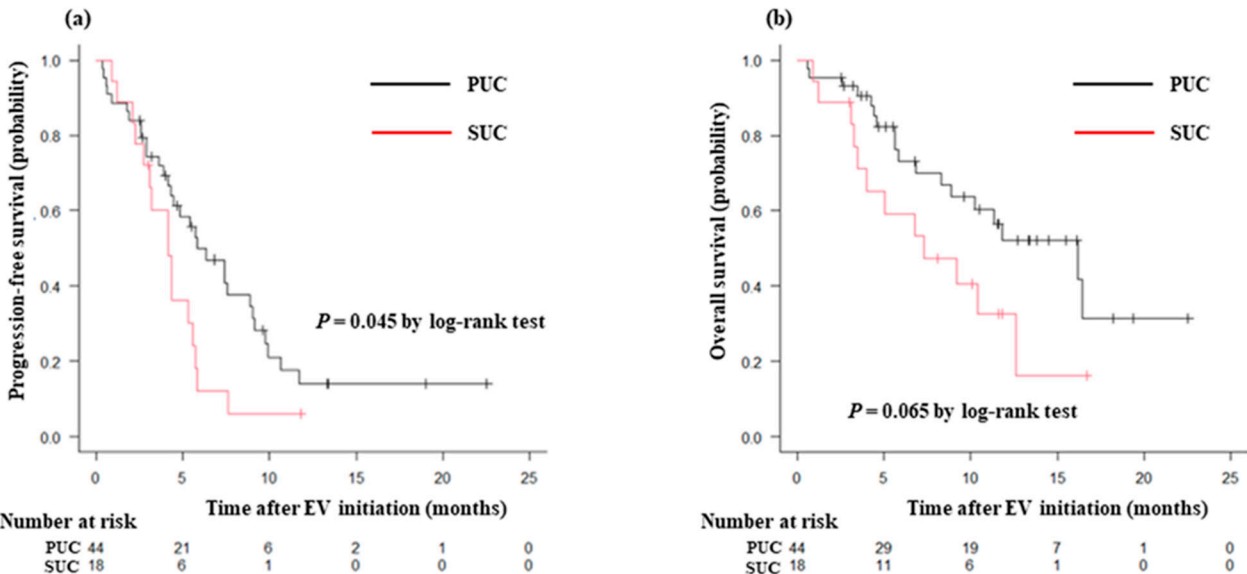

**Figure 2.** Kaplan–Meier curves for (**a**) progression-free survival and (**b**) overall survival after enfortumab vedotin initiation in patients with PUC and SUC. Abbreviations: PUC, pure urothelial carcinoma; SUC, subtype of urothelial carcinoma.

In the univariate and multivariate Cox proportional hazards regression analyses predicting PFS after adjusting for clinicopathological characteristics (Table 4), the presence

of histologic subtype or divergent differentiation was identified as a significant independent predictor of PFS. Additionally, poor ECOG-PS was significantly associated with PFS.

**Table 4.** Results of univariate and multivariate analyses for progression-free survival.

| Variable | Univariate | | Multivariate | |
|---|---|---|---|---|
| | HR (95% CI) | *p* Value | HR (95% CI) | *p* Value |
| Age, years | | | | |
| <73 | 1 | | | |
| ≥73 | 1.21 (0.67–2.16) | 0.529 | | |
| Sex | | | | |
| Male | 1 | | | |
| Female | 0.97 (0.49–1.91) | 0.932 | | |
| ECOG-PS score | | | | |
| 0 | 1 | | 1 | |
| 1 | 1.35 (0.71–2.59) | 0.361 | 1.27 (0.66–2.45) | 0.467 |
| 2 | 2.01 (0.83–4.86) | 0.123 | 1.73 (0.70–4.25) | 0.233 |
| 3 | 3.77 (1.08–13.2) | 0.038 | 4.54 (1.27–16.2) | 0.019 |
| Primary tumor site | | | | |
| Lower urinary tract | 1 | | | |
| Upper urinary tract | 0.83 (0.47–1.48) | 0.527 | | |
| Prior radical surgery | | | | |
| No | 1 | | | |
| Yes | 0.69 (0.38–1.22) | 0.201 | | |
| Anemia | | | | |
| No | 1 | | | |
| Yes | 1.54 (0.83–2.84) | 0.169 | | |
| Liver metastasis | | | | |
| No | 1 | | | |
| Yes | 1.50 (0.75–2.99) | 0.252 | | |
| Prior immune checkpoint blockade | | | | |
| Anti-PD-1 | 1 | | | |
| Anti-PD-L1 | 0.67 (0.36–1.23) | 0.193 | | |
| Histologic type | | | | |
| PUC | 1 | | 1 | |
| SUC | 1.86 (1.01–3.45) | 0.049 | 1.90 (1.01–3.61) | 0.048 |

Abbreviations: ECOG-PS, Eastern Cooperative Oncology Group performance status; PD-1, programmed cell death protein 1; PD-L1, programmed death-ligand 1; PUC, pure urothelial carcinoma; SUC, subtype of urothelial carcinoma; HR, hazard ratio; CI, confidence interval.

## 4. Discussion

To assess the efficacy of EV—an ADC directed against nectin-4—on clinical outcomes according to the histologic types, this study evaluated the therapeutic response to EV monotherapy and its survival outcome in patients with PUC and SUC after platinum-based chemotherapy and ICI therapy. In our cohort, squamous differentiation was the most common subtype element. Although patients with SUC had similar ORR and disease control rate to those with PUC, the duration of response for SUC was short. Patients with SUC had poorer PFS than those with PUC, but no significant difference was observed in OS. Multivariate analysis showed that the presence of histologic subtype or divergent differentiation was an independent predictor of progression after EV initiation. SUC might harbor more aggressive biological features in metastatic disease than in locally advanced disease.

Currently, the efficacy of EV monotherapy as a third-line treatment for patients with advanced UC remains insufficiently investigated in a real-world setting. In the United States, only approximately 3–7% of patients with newly diagnosed metastatic disease

receive third-line therapy [17]. Moreover, the role of EV against SUC has been less explored. Although the EV-301 trial included 15% (*n* = 45) of patients with SUC who received EV, clinical data regarding the response and survival in patients with SUC are still unavailable [5]. The Urothelial Cancer Network to Investigate Therapeutic Experience (UNITE) study has the largest retrospective cohort of patients treated with EV in the United States (*n* = 260), of which 66 patients have SUC [18]. The ORR to EV in patients with PUC and SUC was 58% and 42%, which is consistent with our results. The responses in patients with squamous differentiation (*n* = 28) were CR and PR in one (3.6%) and thirteen (46.4%) patients, respectively. However, compared with our cohort, the UNITE study population included both platinum-pretreated and platinum-naïve patients. In particular, 86 patients (33%) received EV monotherapy as the second or earlier line of treatment [18]. Additionally, response duration and PFS for patients with SUC were not analyzed in the EV-301 trial. Conversely, the duration of response in our PUC cohort is consistent with that in the EV-201 phase 2 trial (median, 7.6 months) [19]. Recently, Zschabitz et al. [20] reported the clinical experience of EV treatment in the largest European study population (*n* = 125); however, only two patients were included in the SUC group.

In our study, the number of other histologic subtypes or divergent differentiation was insufficient. Consequently, we did not include histologic subtypes, such as micropapillary and nested. The heterogeneity in SUC populations may affect the EV therapeutic effect and patient survival. When evaluating the efficacy of platinum-based chemotherapy as the first-line treatment, the duration of PFS or OS in patients with squamous differentiation was comparable to that in patients with other histologic subgroups [9].

Nectin-4 expression is consistently very high in PUC, but prior studies showed lower nectin-4 expression in SUC than in PUC [21]. Although current clinical trials exploring EV in UC do not require nectin-4 expression as a stratifying factor [5], preclinical data have suggested that its expression is necessary for a response to EV [22]. Klümper et al. [23] recently revealed that nectin-4 expression decreased during metastatic spread in 59% of their cases. Interestingly, low nectin-4 expression was associated with reduced PFS duration on EV treatment. In Fan et al.'s immunohistochemical analysis of 161 surgical specimens stained for nectin-4 [24], membranous positive expression was observed in 79.2%, 100%, 65.2%, 21.1%, 78.9%, and 88.9% of conventional UC, squamous, glandular, neuroendocrine differentiation, micropapillary, and nested subtypes, respectively. Moreover, a single-center retrospective analysis on radical cystectomy for SUC showed that sarcomatoid subtypes had low nectin-4 expression [25]. In 2023, Ghali et al. reported the prominent cytoplasmic staining of nectin-4 by a rapid autopsy study [26]. Interestingly, nectin-4 accumulates in the cytoplasm within squamous differentiation or plasmacytoid subtype in bladder cancer, especially within metastatic specimens, but not neuroendocrine differentiation. Currently, changes in nectin-4 expression according to the histologic types during EV treatment remain unevaluated; this evaluation is a major challenge for future studies, including ours. Additionally, in daily practice, the expression of nectin-4 for SUC is difficult to identify in metastatic lesions.

Given that there is still no effective treatment strategy for metastatic SUC because of the exclusion of its analysis from clinical studies, its pharmaceutical therapy is performed according to PUC. Our previous study showed a similar ORR of gemcitabine plus cisplatin (or carboplatin) therapy between the PUC (*n* = 86) and SUC (*n* = 45) groups. However, the SUC group had significantly worse PFS (median, 4.9 months vs. 7.9 months) and OS (median, 10.9 months vs. 18.2 months) than the PUC group [9]. In 2022, we conducted a multicenter retrospective study showing the survival outcomes of pembrolizumab therapy in patients with chemotherapy-resistant SUC [14]. Interestingly, the PFS or OS from the start of pembrolizumab did not significantly differ between the PUC and SUC groups. Therefore, early sequential therapy from platinum-based chemotherapy to ICI therapy may be beneficial for patients with SUC. Evaluating the efficacy of avelumab maintenance therapy or nivolumab adjuvant therapy in patients with SUC is suggested. Moreover,

we expect the EV plus pembrolizumab as a regimen in the ongoing clinical EV 302 trial (NCT04223856) to yield a positive effect for SUC.

ECOG-PS is a well-known prognostic factor concerning chemotherapy response. Patients with poor ECOG-PS (>1) have been excluded from the current clinical trial [27]. In our previous study, the response to EV was observed irrespective of ECOG-PS (proportion of patients with PS score of $\geq 2$ was 26.9%), and the health-related quality of life in patients remained stable from baseline to post-EV introduction [6]. However, the present study suggests that patients with an ECOG-PS score of 3 are poor candidates for EV treatment.

Although surgery was not associated with a better prognosis in this study, previous studies have demonstrated that the treatment of the primary tumor can provide some survival benefit to patients with metastatic UC [28,29]. In the context of improved systemic therapies, aggressive local treatment in well-selected patients could improve the typically poor prognosis of these patients.

This study has certain limitations, such as its retrospective nonrandomized design, small sample size, and short observation period. Histologic assessment of small biopsies potentially has a false-negative risk in the detection of subtype histology, except for prior radical surgery. Given the limited number of patients in our cohort, we could not compare the patients according to the SUC subgroup or primary organ stratified by bladder or upper urinary tract. Moreover, the timing of EV monotherapy was not uniform, with most patients receiving two regimens and the rest receiving three or more before EV introduction. However, EV treatment outcomes depending on histomorphologic groups have not been described in larger populations to date.

Nevertheless, our data suggest that these clinically aggressive histologic types are promising prognosticators for EV treatment regardless of the response. Although multivariate analysis for OS was not estimated, prognostic factors such as late-line treatment should be further explored in the ADC era. The results of response or OS rates for SUC may be attributed to the lack of statistical power. More studies with larger cohorts are needed to further validate our results. Particularly, the difference in the oncological outcomes among subgroups within the SUC group should be evaluated.

## 5. Conclusions

The duration of response and survival after treatment with EV in patients with SUC tended to be shorter than in those with PUC with metastatic disease. However, patients with SUC were few, thus necessitating further investigation.

**Author Contributions:** Conceptualization, A.M. and N.F. (Nobuki Furubayashi); methodology, A.M. and N.F. (Nobuki Furubayashi); formal analysis, A.M. and N.F. (Nobuki Furubayashi); investigation, A.M., N.F. (Nobuki Furubayashi), Y.N., T.T., H.M., Y.S., Y.H., K.K. (Keijiro Kiyoshima) and T.N.; data curation, A.M., N.F. (Nobuki Furubayashi), T.T., H.M., Y.S., Y.H., K.K. (Keijiro Kiyoshima) and T.N.; writing—original draft preparation, A.M.; supervision, M.N. and N.F. (Naohiro Fujimoto); editing, A.M., N.F. (Nobuki Furubayashi) and Y.N.; reviewing, N.F. (Nobuki Furubayashi), Y.N., K.K. (Kentaro Kuroiwa), N.S., I.T. and K.H. All authors have read and agreed to the published version of the manuscript.

**Funding:** This research was funded by JSPS KAKENHI grant number JP23K15773.

**Institutional Review Board Statement:** This study protocol was approved by the University of Occupational and Environmental Health Institutional Review Board (approval no. CRG23-017).

**Informed Consent Statement:** Informed consent was obtained through an opt-out process due to the retrospective study.

**Data Availability Statement:** The data presented in this study are available on request from the corresponding author.

**Conflicts of Interest:** The authors declare no conflicts of interest.

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
