# Peer review of "Prognostic Impact of Histologic Subtype and Divergent Differentiation in Patients with Metastatic Urothelial Carcinoma Treated with Enfortumab Vedotin: A Multicenter Retrospective Study"

_curroncol, doi:10.3390/curroncol31020064_

Round 1

Reviewer 1 Report

Comments and Suggestions for Authors

1) General comments

Minato et al. retrospectively analyzed patients with metastatic upper and lower urinary tract urothelial carcinomas who received enfortumab vedotin (EV) as a third line therapy, and compared the objective response rate (ORR), progression-free survival (PFS), and overall survival (OS) between 44 patients with pure conventional urothelial carcinoma (PUC) and 18 patients with variant urothelial carcinoma (VUC). In multivariate analysis, the presence of subtype histology is one of the independent predictors for shorter PFS. They also indicated that the response duration for patients with VUC was significantly shorter than that for patients with PUC. The manuscript about important issue is generally well-written; however, there are some issues regarding this study that the authors need to clarify.

2) Specific comments

a) Major:

1. The latest World Health Organization (WHO) Classification of Tumours (2022, 5th edition, not 2016, 4th edition) should be utilized. According to WHO 2022, the term “subtype” instead of “variant” is used to express histo-morphological subgroups, as “variant” is frequently used as genetic changes. 

2. Was histological information obtained from only pathological reports without histological review? Additionally, histological diagnoses were considered to be obtained from various types of surgical specimens, such as transurethral resection of bladder tumor (TURBT), cystectomy, nephrouretectomy, and cytoreductive surgery (CRS) for metastatic lesions. This information should be included in the manuscript and Table 2. As a nature of partial resection, histological assessment of TURBT and CRS potentially has a false-negative-risk for detecting subtype histology, which should be added as a limitation of the study.

3. Lines 196-197 and Lines 256-258; As authors described, the present data is insufficient to indicate the favorable efficacy of EV in patients with urothelial carcinoma with squamous differentiation. This description should be at least excluded from the conclusion.

b) Minor:

1. Line 32: ORR was not significantly different between patients with PUC and subtype histology, which is confusing because of the following description, “As for OS, no significant difference was observed”. Please accurately describe the data.

2. Line 35: “shorter PFS” instead of “PFS”.

Reviewer 2 Report

Comments and Suggestions for Authors

Dear Authors,

Please find attached my comments and suggestions for your paper.

Best of luck with your future research!

Reviewer 3 Report

Comments and Suggestions for Authors

The study is interesting because it investigates the effectiveness of additional lines of systemic therapy for patients who generally have few therapeutic options and a poor prognosis, and for whom systemic therapies have made strides in recent years. Overall, the study is well-written.

The main limitation, already acknowledged by the authors, is related to the small sample size, also due to the difficulty in enrolling these patients. However, the authors should explicitly acknowledge that the lack of statistical significance for some analyses is likely due to this limitation.

Please provide more details about the therapy protocol: in case of no progression, unacceptable adverse events, or consent withdrawal, how many cycles were planned?

Two additional columns (lower vs upper) to present histology per primary tumor site in Table 1 would be helpful.

The authors should specify how the metastatic status was assessed. Did any of the patients undergo metastasectomy?

Although surgery did not show an association with better prognosis in this study, other studies have demonstrated that the treatment of the primary tumor can provide some survival benefit to patients with metastatic urothelial carcinoma. In the context of improved systemic therapies, aggressive combined therapy in well selected patients could improve the typically poor prognosis of these patients. In this regard, the following studies should be cited:
10.1016/j.urolonc.2021.01.031
10.1200/JCO.2016.66.7352

Comments on the Quality of English Language

Good quality

Round 2

Reviewer 1 Report

Comments and Suggestions for Authors

The manuscript has been revised well and is now acceptable for publication.

Reviewer 2 Report

Comments and Suggestions for Authors

Thank you for going through my comments and taking the time to provide further clarifications. I am happy with the changes you have made.

Best of luck with your future work!